# What do doctors understand by spiritual health? A survey of UK general practitioners

Orla Whitehead , Carol Jagger, Barbara Hanratty

## ABSTRACT

**Background**  In the UK, doctors' regulatory and professional bodies require general practitioners (GPs) to consider discussing spiritual health as part of the consultation. However, spiritual health is not defined in guidance, and it is unknown what individual doctors understand by the term.

**Research question**  What do GPs understand by the term 'spiritual health'?

**Aim**  To explore how GPs understand and define spiritual health.

**Design and setting**  Survey of GPs in England 9 April 2019–21 May 2019.

**Method**  A mixed-methods online survey asked practising GPs in England qualitative free text questions—'What does the term 'Spiritual Health' mean to you?' and 'Any comments?' after five vignettes about discussing spiritual health with patients. These were subject to thematic analysis using a priori themes from the literature on GP definitions of spiritual health, and on attitudes towards the topic.

**Participants**  177 practising GPs in England.

**Results**  177 GPs responded to the survey. Understanding of spiritual health fitted into three themes: self-actualisation and meaning, transcendence and relationships beyond the self, and expressions of spirituality. A full range of views were expressed, from a minority who challenged their role in spiritual health, through to others enthusiastic about its place in healthcare.

**Conclusion**  Spirituality and religiosity are understood by English GPs to be distinct concepts. A consensus definition of spiritual health incorporating the themes identified by working doctors, may be helpful to support GPs to follow the recommended guidance in their practice.

## Strengths and limitations of this study

► Spiritual health is an undefined concept in guidance for general practitioners (GPs), but UK doctors have it within their professional remit. This survey offers a threefold definition of spiritual health developed by family doctors.

► This is the largest survey of doctors in Europe on the topic of spiritual health, as 177 practising GPs responded, a large number from a group who are often difficult to engage in research. A broad range of views was expressed, suggesting that selection bias was minimal.

► Data were collected on religious views, despite this being a potentially sensitive topic.

Population Health Sciences Institute, Newcastle University, Newcastle upon Tyne, UK

**Correspondence to**
Dr Orla Whitehead; orla.whitehead@newcastle.ac.uk

## INTRODUCTION

Modern medicine may have little time for the spiritual aspects of life, however, doctors have always been required to address more than physical problems.[1] Within the National Health Service, interest in holistic care and mental well-being is growing.[2] Holism underpins primary care, and general practitioners (GPs) are expected to treat patients within their social and cultural context.[3] The UK regulatory body, the General Medical Council (GMC), expects doctors to consider

the religious, social and cultural aspects of a patient's problem,[4] and the Royal College of General Practitioners (RCGP) includes spiritual health in their curriculum for GP trainees.[3] National Institute for Health and Care Excellence guidance also recommends that discussion of spiritual beliefs should be part of the care of dying patients.[5] A recent article has advocated for an 'embedded' model of spiritual care within UK general practice.[6] Discussing spiritual health may be a beneficial intervention for patients,[7] and there has been speculation that it could increase job satisfaction for GPs.[8] Spirituality can also be a resource for health and well-being. Religiosity and spirituality are associated with longevity,[9 10] and unmet spiritual needs may be detrimental to health.[11] Research into interactions between spirituality and health is a growing field.[9 12] In recent years, arguments have been made for the importance of embedding spiritual health within general practice.[6] However, the interface between spirituality and medicine can be controversial.[13] Pathological spiritual and religious beliefs may develop, and some UK GPs who shared their faith with patients have been investigated by the GMC.[14]

> **Box 1  Definitions of spiritual health used in research**
>
> ► Humanity—spirituality, the spirit or the soul is part of the universal human experience.[20 39]
> ► Meaning and purpose, or self-actualisation.[17 34 39–42]
> ► Inner strength, peace and resilience.[40]
> ► Transcendence, or a sense of connectivity to aspects outside the self, for example, with the community, society, family and nature.[17 39 41 42]
> ► Relationship with the divine or sacred.[39]
> ► Expression—beliefs, values, traditions, rituals and practices.[20 41]

There is no consensus or widely accepted definition of spiritual health, despite its proposed place in GPs' work. Spirituality means different things to different people, and understanding may shift across the life course.[15] Some definitions invoke the symptomatology of mental illness, which causes confusion and negative attitudes to the concept.[16 17] Danish authors have argued that the term 'existential health' is more appropriate for a secular multicultural society,[18] while others see existential and spiritual health as synonymous.[19] Similarly, spirituality and religiosity are often conflated.[20–24] For some, spirituality is personality trait predisposing to concern with the metaphysical[25] that has played an important role in human societies and medicine.[26] Previous quantitative survey work with GP trainers in Scotland found that spirituality was perceived by educators as a meaningful, useful, but unclear concept.[27] Researchers investigating how spiritual health should be discussed in primary care have adopted a variety of definitions drawing themes from traditional religion through to humanism (box 1). One of the most frequently adopted is a consensus definition from international holistic care conferences that encompasses all the concepts listed in box 1, apart from inner peace, strength and resilience.[27–29] However, while these definitions may be useful to those interested in the topic, or within academia, it is not known what working definition is being used by GPs in a clinical context.

A clear understanding of spiritual health will be essential if it is to become a routine part of holistic care in general practice. This study aims to explore how GPs understand and define spiritual health in their day to day practice, and build the evidence base regarding GP attitudes towards the topic.

## METHOD
### Design
A mixed-methods online survey was developed and used to explore GP definitions of the term 'spiritual health', attitudes towards the topic and views on discussing spiritual health with their patients. (Data on the use of structured tools to support discussions within consultations are reported elsewhere). Information collected from the participants included demographic characteristics (sex, ethnicity and religion); geographical region and years of work experience.

Participants were asked to rate themselves on a five-point Likert scale, in response to two statements: (1) I am a spiritual person and (2) I am a religious person. Spiritual health was deliberately undefined in the survey; GPs were asked 'what does the term spiritual health mean to you?' and invited to share any other thoughts on this topic, prompted by five patient vignettes.

The survey was sent to qualified GPs in England who were working in standard general practice settings, out of hours and urgent care. Included were all practising GPs in England, people who were temporarily out of practice (eg, maternity leave) were encouraged to participate, but those who had retired and would not return to practice were excluded. The survey was sent via Clinical Commissioning Group (CCG) lists, CCG newsletters, professional groups and practice managers in Spring/Summer 2019, all 211 CCGs were approached, both convenience and snowball sampling methods were used.

### Data analysis
Free-text responses to the question 'what does the term spiritual health mean to you', were analysed using a deductive thematic analysis, based on a priori themes developed by immersion in the literature (box 1). The analysis process followed a standard path: reading and rereading of the data, coding based on key words and ideas expressed by respondents, comparing and contrasting of similar codes, aggregation into topics that were then grouped into the a priori themes and any other additional themes.[30] These themes were reviewed to develop the central concepts within the data. Outlying cases were examined, to identify insights from respondents most and least comfortable with the topic. Qualitative software was not used to manage these data, rather the concepts were manipulated using coloured pens, paper and scissors. The researchers were cognisant of their own identities, cultures and religious backgrounds in their approach to the topic. Feedback was sought from GPs from other backgrounds in the survey development. Authors 1 and 3 are practising GPs.

A similar process was followed to analyse the free text provided in the space for additional comments. In this case, GP responses were also mapped against the attitudes to spiritual care identified in Appleby *et al*'s quantitative work with Scottish GP trainers. They identified four categories of attitudes labelled as rejecting, guarded, pragmatic and embracing.[27]

### Patient and public involvement
Patient and public involvement was important to the development of the research and analysis, and the authors wish to acknowledge, and are grateful for, the contribution of Voice North. The group had diverse opinions on the topic, but all recognised the need for

**Table 1** Characteristics of respondents and self reported spirituality and religiosity

| | No (%) (n=177) |
|---|---|
| **Sex** | |
| Male | 65 (37) |
| Female | 111 (63) |
| **Ethnic group** | |
| White British | 139 (79) |
| Any other White background | 7 (4) |
| White Irish | 5 (3) |
| Indian | 6 (3) |
| Any other mixed/multiple ethnic background | 4 (2) |
| Other background | 15 (9) |
| **Religion** | |
| Christian | 110 (63) |
| Other | 12 (7) |
| None | 49 (28) |
| **Country of primary medical qualification** | |
| England | 144 (81) |
| Scotland | 14 (8) |
| Elsewhere in Europe | 8 (5) |
| Asia | 6 (3) |
| Africa or Americas | 5 (3) |
| **Country of GP training** | |
| England | 168 (95) |
| Scotland | 5 (3) |
| Other | 4 (2) |
| **I am a spiritual person*** | |
| Agree | 122 (69) |
| Disagree | 28 (16) |
| No opinion | 27 (15) |
| **I am a religious person*** | |
| Agree | 85 (47) |
| Disagree | 67 (39) |
| No opinion | 25 (14) |

*Responses to a five-point Likert scale, with strongly agree/disagree and agree/disagree aggregated.
GP, general practitioner.

holistic care, and the utility of a working definition of 'spiritual health'.

## RESULTS

One hundred and seventy-seven practising English GPs responded to the survey, shown in table 1. The majority were female, of white British origin, and with a primary medical qualification obtained in the UK. Most respondents (69%) considered themselves to be spiritual people.

Just under half (48%) considered themselves religious, 70% of respondents reported that they had a religion (63% Christian, 7% named other religions). One-fifth of respondents (21%) reported that they were spiritual, not religious. All respondents who agreed they were religious also agreed they were spiritual people.

One hundred and fifty-seven out of 177 participants provided a free-text response to the question 'what does the term spiritual health mean to you?' These responses, ranging in length from 1 to 105 words, were subject to a thematic analysis. Most of the definitions and understanding of spiritual health could be coded into the three a priori themes: (1) self-actualisation and meaning, (2) transcendence and relationships beyond the self and (3) expressions of spirituality. One additional theme was developed and labelled (4) 'meaningless'. Religious participation cut across all three of these themes; development of the self is included in many world religions, as is transcendence, community and relationships to other humans, the world, and/or the divine, and of course expression of beliefs. Only one respondent used the term existential health, and a small number of respondents commented that spiritual health is synonymous with mental health. In the description of themes below, direct quotations from participants are included within the text without attribution.

### Self-actualisation and meaning

Self-actualisation and meaning brought together ideas around humanity/human experience, a sense of spirit or soul, inner strength, peace and resilience and meaning, purpose or self-actualisation. Acceptance of the self, the situation, with a sense of peace and meaning to life was mentioned, both with and without reference to God. Participants referred to the health of a soul, or inner self, which could be either peaceful or disrupted with concerns, conflict or guilt. A sense of purpose was viewed as giving fulfilment and meaning to life, or as an understanding of the identity: 'Understanding where I have come from and where I am heading.' Personal resilience and inner strength were mentioned as a healthy outcome of acceptance of the soul, of our humanity, and of our status as 'finite beings.' One participant mentioned 'security in who we are, and why we're here', expressing that sense of personal identity and purpose that appears to be part of the human experience. Looking into the self, and something of the individual that is 'neither physical nor mental', and finding peace and meaning there, is core to this concept. However, few participants mentioned ideas without going on to refer to the position of the individual and their self-actualisation within the world/universe, 'understanding your own place and role within society.'

### Transcendence and relationship beyond the self

This theme encompassed ideas of connections to community, nature and family, with relationship with the divine and sacred. A majority of participants mentioned relationship and connections to the community, 'something

bigger', family and the world/nature, 'Being in tune with people. Enjoying and respecting the natural world.' Participants referred to: 'believing in something bigger than just you,' and 'connectedness to the world they can't see.' Participants were often keen to be inclusive in their definitions, referring to 'being comfortable with the world around you and any faith entity you believe in', or referring to 'a higher being' rather than a specific deity. Some participants were clear that, for them, this was part of an accepted religious framework, for example, 'a relationship with Jesus' (one participant) or 'myself as a Child of God' (18 participants mentioning God). Looking outwards, 'the state of one's capacity for love' and a belonging to something outside the self, appears to underpin this concept.

### Expression of spirituality

The concepts of spirituality of the self, and of our connections outside the self, come together into expressions of spirituality, such as values, rituals, practices and lifestyles. One participant stated 'it embraces everything from mindfulness to religious practice (short of pathological religious delusion.)' Many participants mentioned having 'faith', although what this was faith in was not always expanded on. The freedom to be 'able and … visibly practice my faith' was mentioned by one participant, suggesting that spiritual health requires freedom of expression to be permitted. The use of morals, and personal ethics to influence behaviour, for example, 'believing in something abstract that guides behaviour and thinking', was mentioned as part of spiritual health. The expression of these values and morals was often linked to membership and belonging to a community, or the seeking of self-actualisation.

### Spiritual health as a meaningless term

A small minority of respondents (6%) reported that spiritual health conveyed no meaning, or was unclear, to them, but there was little elaboration on this assertion.

### Attitudes to spiritual care

Respondents were invited to add any other thoughts on spiritual health and care in general through unprompted free text boxes asking 'any comments'. One-third of participants contributed their views, and data relating to understanding and definition of spiritual health were mapped to an existing framework developed by Appleby *et al* in table 2. This is derived from a small qualitative survey of 19 Scottish GPs[31] which was then used by the authors with a larger group of Scottish GPs in a quantitative study,[27] and presents a categorisation of the spectrum of attitudes to spiritual care already used in a UK context. The perceived need for cues by a large number of respondents, and the low numbers of respondents who fit into the 'rejecting' attitude, would suggest that most respondents fall in to the 'guarded' and/or 'pragmatic' attitudes identified.

## DISCUSSION

This large study of practising English GPs suggests that spiritual health has a meaning for many doctors, but that this meaning is distinct from religion. Three themes were identified to encompass GPs' understanding of spiritual health: self-actualisation and meaning, transcendence and relationships beyond the self, and expression of spirituality. Difficulty in articulating exactly what is conveyed by the term spiritual health was a common challenge.

### Comparison with other work

We have been able to present an in-depth and nuanced description of GPs' understanding of spiritual health, with a study that is larger than previous work in the UK. We identified some views of spiritual health that were similar to those expressed in interviews with 19 Scottish GPs and a quantitative survey of 87 GP trainers, including a lack of meaning of the term spiritual, links with humanity and the divine.[27 31] The breadth of views in our study was similar, enabling us to map responses to a framework based on Appleby *et al*'s four descriptors of attitudes to spiritual care, as shown in table 2. Other work has focused on a priori definitions of spiritual health,[28 32–34] and while Puchalski *et al*'s consensus definition[29] gives a useful expert definition, it was previously unknown what definition English GPs were using in their work. The definitions given by English GPs reflects those given in the literature (box 1). While some authors describe religion and spirituality as interchangeable,[35] one-fifth of the GP participants here described themselves as spiritual but not as religious, showing they believe these are different concepts. Most other literature originates from the USA, which may promote a definition that is appropriate only to that sociocultural context, whereas building on Appleby *et al* allows the UK evidence base to be developed. Assing Hvidt *et al*'s work on 'existential health' in Denmark[36] is an interesting comparator to this work, however their definition of existential health differs from that of 'spiritual health' here by English GPs, possibly due to cultural differences between the UK and Denmark.

### Limitations

While the response rate was good for this sort of survey, it was a small proportion of GPs in England, and there could be sampling bias with those with an interest in the topic being more likely to respond. It is not known whether the high rate of identification as a 'spiritual person' is due to this being the case in the wider GP population, or just those sampled. Unfortunately, the sample was not representative of GPs with non-white backgrounds.

Respondents differed from the UK GP population registered with the GMC, which should be considered alongside the findings. Compared with the GP population registered with the GMC, respondents were more likely to be female (63% vs 53%) and to have a medical qualification from the UK (89% vs 79%), but less likely to be of black or minority ethnic origin (14% vs 25%).[37]

**Table 2** Attitudes to spiritual care: GP views mapped onto a framework of enthusiasm for spiritual care

| Categories of attitudes to spiritual care among GPs identified in a quantitative survey by Appleby et al[27] | Example of attitude | Views expressed by GPs in this survey |
|---|---|---|
| Rejecting | Discomfort with the concept and any spiritual healthcare | "Not comfortable including spirituality in consultations." Male, GP for 20–30 years<br>"What can I say. There is no evidence for any God." Male, no religion, GP for 20–30 years<br>"I don't feel this has anything to do with my role as a GP." Female, no religion, GP for 6–10 years<br>"I'm not sure we need to be the people to ask about spiritual things. This sits better with a religious leader. I don't feel asking these questions really helps me to help them. It doesn't sit comfortably with a GP role in my opinion." Female, Christian, GP for 20–30 years |
| Guarded | Limited circumstances where a GP should be involved | "I think I am very hesitant in asking as you hear of doctors being accused of causing offence etc." Female, Christian, GP for 6–10 years<br>"I don't see my role to take over a spiritual conversation unless directly relevant to medical choices." Female, no religion, GP for 20–30 years<br>"I'm not sure this is my place and the only discussion I might have would be at the end of life." Female, no religion, GP for 6–10 years<br>"Signposting to appropriate people is appropriate and probably a better use of the patient's time." Male, no religion, GP for 6–10 years<br>"Unless patient brings it up or it's an end of life planning content it's none of my business." Female, Christian, GP for 6–10 years |
| Pragmatic | Willing to address the topic at patient request | "I can remember the first time I asked a palliative patient about whether they had anyone they discussed spiritual matters with, I was nervous about asking, but the amazing response has made me ask almost all of my palliative patients since." Male, no religion, GP for 11–20 years<br>"I will NEVER bring it up but would engage professionally with the patient in a positive manner." Male, no religion, GP for 20–30 years<br>"I feel comfortable discussing spirituality if led by the patient." Female, Christian, GP for 0–5 years<br>"The apparent importance of the spiritual aspects in Derek's life and pending death would encourage to broach this more explicitly and respond accordingly." Male, Christian, GP for 20–30 years |
| Embracing | Enthusiasm for the topic, and acceptance of the importance in healthcare | "By comfortable I mean that I'm not afraid to discuss these things and think it's important." Female, Christian, GP for 11–20 years<br>"I would be comfortable asking what he is reading in the bible and discussing passages that may bring comfort to him and his wife if they wanted." Female, Christian, GP for 20–30 years<br>"They have hugely valued talking about faith and praying together as we share the same faith. I have been able to still explore faith with those of different faiths." Male, Christian, GP for 11–12 years |

GP, general practitioner.

Recruiting GPs via CCG newsletters and other forms of publicity resulted in an good sample size, but provides no information on people who decided not to take part. The range of views that were expressed provides some confidence in our approach, but we acknowledge that people with strong views from either the 'rejecting' or 'embracing' ends of the spectrum may be more likely to have participated. Respondents were drawn from across England, with a majority in the north (63%). This was probably due to greater engagement with the project from CCGs in the area local to the research team.

Respondents differed from the UK GP population registered with the GMC, which should be considered alongside the findings. Compared with the GP population

registered with the GMC, respondents were more likely to be female (63% vs 53%) and to have a medical qualification from the UK (89% vs 79%), but less likely to be of black or minority ethnic origin (14% vs 25%).[37]

## Implications for research and practice

Spiritual health proved to be a divisive topic that evoked strong opinions. The GMC and the RCGP expect GPs to include spiritual health in care, but this research exposes heterogeneity in GPs' understanding of the topic. Whether, and how, an individual doctor's understanding of spiritual health influences their practice, is unknown. It is possible that discussion of spiritual health leads to enhanced patient outcomes, in areas such as end of life care. The positive, negative and unintended, consequences of introducing discussion of spiritual health into consultations requires further study. Busy GPs are likely to require evidence of benefit before they would routinely widen the scope of their existing practice. Impact on workload is an important area for future scrutiny. GPs are not well placed to provide spiritual care,[34] and how to ensure people with spiritual needs are directed to appropriate support requires further investigation. Social prescribing and primary care chaplains offer potential routes to spiritual support for patients,[38] and both may help to reduce demands on doctors in primary care.

## CONCLUSIONS

Views on spiritual health among GPs are heterogeneous. A clear definition would facilitate future research, and may help to promote discussion in the consultation. The themes identified in this study are generated from working GPs and could form the basis of a socioculturally sensitive and encompassing definition of spiritual care. Without a shared understanding, holistic assessment of patients' needs and referral for appropriate support will be an ongoing challenge.

**Acknowledgements** Thank you to all the GPs who participated, and all those in CCGs and CRNs who helped with recruitment. Thank you to Voice North members who helped shape the study and analysis.

**Contributors** OW conceived the study, with input from BH and CJ. OW developed the study design with input from CJ and BH. OW developed the survey, with input from BH and CJ. OW performed the analysis and wrote the first and successive drafts of the manuscript. OW, CJ and BH interpreted the data, critically revised the manuscript for important intellectual content and approved the final version of the manuscript.

**Funding** OW was funded by a post-CCT GP Fellowship funded by Health Education North East and Durham Dales and Easington Clinical Commissioning Group, and funds from the National Institute for Health Research (NIHR) School for Primary Care Research. Grant reference HEE REF 0150/8116. BH was part funded by the North East and North Cumbria Applied Research Collaboration.

**Disclaimer** The views expressed are those of the author(s) and not necessarily those of DDES CCG, the NIHR or the Department of Health and Social Care.

**Competing interests** None declared.

**Patient and public involvement** Patients and/or the public were involved in the design, or conduct, or reporting, or dissemination plans of this research. Refer to the Methods section for further details.

**Patient consent for publication** Not required.

**Ethics approval** Ethics approval was sought and obtained from Newcastle University on 27 February 2019.

**Provenance and peer review** Not commissioned; externally peer reviewed.

**Data availability statement** Data are available on reasonable request. Data are saved on Newcastle University secure servers and may be available in negotiation with the first author.

**ORCID iDs**
Orla Whitehead http://orcid.org/0000-0002-4171-8583
Barbara Hanratty http://orcid.org/0000-0002-3122-7190

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
