## [Reviewer comments · BMJ Open]

ARTICLE DETAILS

TITLE (PROVISIONAL)	What Do Doctors Understand By 'Spiritual Health'? A survey of UK general practitioners
AUTHORS	Whitehead, Orla; Jagger, Carol; Hanratty, Barbara

VERSION 1 – REVIEW

REVIEWER	Melanie Rogers University of Huddersfield
REVIEW RETURNED	19-Jan-2021

GENERAL COMMENTS	- I am unsure what is meant by spiritual therapies in the key words. I would suggest removing this and adding spirituality - Abstract, please also include research question, rephrase design. Identify that this is a qual study investigating practicing GPs understanding of spiritual health. In setting how long was the survey open for? Results in abstract focus on GPs own spirituality, if this is the case the research question and objective should be different. How did understanding of spiritual health fit with their own beliefs. In conclusion suggest change to spirituality and religion rather than spiritual health and religiosity. Strength/Limitations- spiritual health is not an undefined concept. Does a limitation include a bias in response of around 80% who define themselves as spiritual or religious Introduction- I would omit use of word healer so early, focus on the drivers for integrating spirituality into primary care, definitions and literature available. Second point why only focus on secular countries, if doing this define why England and why has moved towards secularism. Design- The authors states this is a qual study but they ask 3 quant questions so mixed methods. Please identify how was the tool developed? I am unsure why the bracketed statement re tools to discuss consultations is here. Why were the 3 quant questions asked and how do they link to the research question? Abstract reporting differs to design aim. Where is reporting of ethical approval. Analysis- Which methodology did you utilise? unclear from design if only 1 qual question was asked. What does the author mean by deviant, explain. Justify manual coding. How did the researchers limit their bias? Results- I think there needs to be clarity as to why the GPs were asked primarily about whether they were spiritual/religious or
--

	spiritually healthy. Rephrase line 50, of the 85 respondents xxxx How were the a priori themes developed, expand please for clarity. How many participants said dread of the afterlife- line 38, p8, if an outlier response please identify this. Justify theme of meaningless - only 1 line included. In general the themes could benefit from some revision. When comments are made for example transcendence "child of God" identify if these are minority comments p10 line 50, why were data mapped to appleby, expand Be consistent about UK GPs as abstract states England Discussion line 15-26, pg 23 would be better placed in limitations section. Line 29 what does this mean? The discussion is very weak. Why is Ellis et all (see below) not discussed. Table 2 could be utilised much better. line 6, p9 when saying a majority of participants please add the number/percentage. line 36, p9 1 participant (remove s) Ellis M, Vinson D & Ewigman B (1999) Addressing Spiritual Concerns of Patients: Family Physicians' Attitudes and Practices. Journal of Family Practice 48 105-9 Ellis M, Campbell J, Detwiller-Breidenbach A & Hubbard D (2002) What do Family Physicians think about Spirituality in Clinical Practice? The Journal of Family Practice 51 (3) 249-254 Ellis M & Campbell J (2004) Patient's Views about Discussing Spiritual Issues with Primary Care Physicians. Southern Medical Journal 97 (12) 1158-1163 Discussion on beneficial intervention should include some of Koenig' findings re positive impacts on health rather than initial link to job satisfaction. If adding this why is it linked?
--	--

REVIEWER	Maryam Rassouli Shahid Beheshti University of Medical Sciences
REVIEW RETURNED	08-Feb-2021

GENERAL COMMENTS	The manuscript has been written about an important topic which should be considered as a priority in providing health care. However, there are some concerns in this regard which should be considered as follows: Introduction Please mention the exact aim of the research at the end of the introduction section. Methods The section needs more clarification. How was sampling method? How was the sample size calculated? Is there any inclusion/exclusion criteria for entering GPs in the survey?
--

	The content of the survey and its structure is not clear. There was just a unique question regarding meaning of spiritual health, which participants should answer. It seems that the research approach is a qualitative one so I'm wondering why researchers didn't use interviews as well as focus group discussions in order to gathering data. According to page 3, lines 56-57, data analyzed based on the provided definitions in the box 1. Therefore, it seems that the researchers used "directed content analysis". Please explain which approach has been used for analyzing data according to this type of content analysis. It is recommended to mention what researchers did for confirming the trustworthiness of data. The reason for bringing the separated paragraph in page 4, line 28 (patient & public involvement) is not clear. It is recommended to describe number of participants who didn't answer the survey. It seems that the research has two parts that if the results of both parts are presented together, it can provide valuable information to readers. Due to the fact that the qualitative part was done only through questionnaires without interviewing with participants, it does not provide much information and therefore the discussion of the findings does not add anything to the existing knowledge.
--	--

REVIEWER	Dirk Labuschagne Rush University Medical Center, Supportive Oncology
REVIEW RETURNED	16-Feb-2021

GENERAL COMMENTS	This paper reports UK general practitioners' (GPs) understanding of spiritual health. Regulatory bodies in the UK require GPs to consider their patients' spiritual and religious needs. Yet, challenges around doing so and having no consensus on an accepted definition for spiritual health complicate this. The authors seek to explore GPs definitions of spiritual health as a first step in this regard. The attempt to do so is well-written, with a comprehensive introduction and engagement with the literature, a clear methodology and proper summary of the results. There are a few suggestions/questions to further improve this study. p. 0, line 35: This reviewer's reading of the study results doesn't agree with this statement that the GPs were asked about discussing spiritual health with patients. Another aspect of the survey may have asked about it (results not reported in this paper) but we read results asking GPs about their understanding of spiritual health. If this additional aim is part of the study (especially in regard to this reviewer's comments below re: p. 4, line 17), then this should be included in the aim and discussion sections. p. 0, line 45: Participants reported here belong in the result section; participants may just be qualified GPs in England (see p. 3, line 40) p. 3 line 9-10: Authors emphasize need for clear understanding of spiritual health and state aim of study as to "address this gap." It is not clear how reporting the GPs understanding of spiritual health is filling this gap, since similar to the wide variety of definitions already present in the literature (p. 2, line 58), GPs would probably have a wide variety of views, not clearly defined. At the very least, it seems to test whether GPs have a similar wide variety of definitions, thereby underlying the need for a consensus definition.
--

	p. 4, line 17: How did the free text space for additional comments result in the data that is presented in Table 2, p. 13? The data in Table 2 appears to come from a specific prompt, perhaps about their views regarding discussing spiritual health with patients? If so, does this necessitate adding this as a secondary aim of this study and updating the discussion, bringing into dialogue Table 2's findings with the relevant literature? p. 6, line 48-52: With the identification of "Self-Actualization and Meaning" as a theme, "Meaningless" isn't the best description here since it may be misleading. It denotes spiritual health as potentially being viewed as senseless or futile, rather than the more practical understanding that for some GPs spiritual health felt unclear or undefined. p. 8, line 7: It is not clear to this reviewer based on the study results how the meaning connected to spiritual health was distinct from religion. The authors didn't define their working definition for religion, and in the themes identified several distinctly religious items emerged, such as punishment, afterlife, soul, Jesus, and child of God. p. 8, line 46: It is not a clear parallel to connect the lack of meaning as identified in the referenced study (where GPs were asked about their understanding of spirituality) and the present study where GPs left as undefined their definition for spiritual health. p. 15, line 13: The authors' first names are listed, correct to Cobb, Puchalski and Rumbold
--	---

VERSION 1 – AUTHOR RESPONSE

Reviewer: 1

Dr. Melanie Rogers, University of Huddersfield

Comments to the Author:

- I am unsure what is meant by spiritual therapies in the key words. I would suggest removing this and adding spirituality

Spiritual therapies is a MESH heading which includes "Mystical, religious, or spiritual practices performed for health benefit." As part of the third 'theme' in the paper "expressions of spirituality", it was felt that this would include spiritual and/or religious practices performed for health benefit. Spirituality was also listed separately.

- Abstract, please also include research question

Thank you, we would want to ensure the research question was clear, and so we have added:

"Research question:

What do GPs understand by the term 'spiritual health'."

rephrase design. Identify that this is a qual study investigating practicing GPs understanding of spiritual health. In setting how long was the survey open for?

The abstract has been rewritten for clarity as advised:

"Design and setting: Survey of GPs in England 9th April-21st May 2019

Method

A mixed methods online survey asked practicing GPs in England qualitative free text questions-

"What does the term 'Spiritual Health' mean to you?" and "Any comments?" after five vignettes about discussing spiritual health with patients. These were subject to thematic analysis using a priori themes from the literature on GP definitions of spiritual health, and on attitudes towards the topic."

Results in abstract focus on GPs own spirituality, if this is the case the research question and objective should be different. How did understanding of spiritual health fit with their own beliefs. In conclusion suggest change to spirituality and religion rather than spiritual health and religiosity.

Thank you, there was a great deal of data in this study, and we wish to communicate the key findings clearly, in response to the research question of how GPs define the term 'spiritual health'. The results section of the abstract was re-written to clarify: "177 GPs responded to the survey. Understanding of spiritual health fitted into three themes: self-actualisation and meaning, transcendence and relationships beyond the self, and expressions of spirituality. A full range of views were expressed, from a minority who challenged their role in spiritual health, through to others enthusiastic about its place in health care."

The conclusion of the abstract was also rewritten to "Spirituality and religiosity are understood by English GPs to be distinct concepts."

The term spiritual health is used to reflect the language of the World Health Organisation and the RCGP when they discuss holistic health. The term spirituality can be quite broad in scope, and to enable pragmatic health focussed research, without detours into philosophy, the term spiritual health is used. While the participants were asked "What is your religion?" (i.e. identity with a religion, with the purpose of checking diversity of sample, and for other parts of the study), the statement here is about the number of GPs answering "I am a religious person" (i.e. religiosity rather than identifying with a religion) compared with answering "I am a spiritual person" (asking about spirituality). Therefore, the terms 'spiritual health' and 'religiosity' are used purposefully, as distinct from where 'spirituality' and 'religion' are used.

Strength/Limitations- spiritual health is not an undefined concept.

Within GP guidance, and the GP training curriculum, spiritual health is referred to, but undefined. The authors are aware of definitions offered internationally, and these were used in developing the a priori themes for analysis. Therefore, this was rewritten to "Spiritual health is an undefined concept in guidance for GPs, but UK doctors have it within their professional remit. This survey offers a threefold definition of spiritual health developed by family doctors."

This has been expanded in the introduction, to explain "However, while these definitions may be useful to those interested in the topic, or within academia, it is not known what working definition is being used by GPs in a clinical context."

Does a limitation include a bias in response of around 80% who define themselves as spiritual or religious

It is unclear as to whether this is a strength or limitation. In our study, 69% of participants defined themselves as spiritual people, with 48% describing themselves as a religious person. Since this is the first survey of this kind, it's not known whether this is a limitation, or reflective of the population, and while a greater proportion of the population stated they had a religion back in the 2011 census, it's not known whether that is the same as being 'a religious person', or religion as part of identity. While some argue that spirituality is a universal human experience, and that there are few cultures without some spiritual ideology, it is interesting that 31% of the sample would not describe themselves as spiritual people. It is not known what the population levels are of self-identification as a spiritual person amongst GPs to know if this is a limitation, or a strength, since this was the first study known to ask this question. Therefore, the text hasn't been added to as to whether this is a strength or limitation of this study.

Introduction- I would omit use of word healer so early,

The text has been revised to: "Modern medicine may have little time for the spiritual aspects of life, however, doctors have always been required to address more than physical problems."

focus on the drivers for integrating spirituality into primary care, definitions and literature available.

There has been recent literature in the BJGP on this topic, arguing for greater inclusion of spiritual health and wellbeing within primary care, which prompted this study, for example: Bornet et al: Spiritual care is stagnating in general practice: the need to move towards an embedded model British Journal of General Practice 2019; 69 (678): 40-41. DOI: 10.3399/bjgp19X700613 and Hamilton et al: Should GPs provide spiritual care? British Journal of General Practice 2017; 67 (665): 573-574. DOI: 10.3399/bjgp17X693845 The benefits of holistic assessment are accepted by GPs in England, and it is included in the RCGP curriculum for GP trainees, the purpose of this study is not to argue for more inclusion of the topic, but, given it is supposed to be included in primary care, to explore what the term 'spiritual health' means to GPs in practice.

The text "A recent article has advocated for an 'embedded' model of spiritual care within UK general practice.²" referring to Bornet et al has been added to the introduction.

Second point why only focus on secular countries, if doing this define why England and why has moved towards secularism.

This study was with GPs in England, which does appear to be moving towards secularism, with the numbers of those answering 'no religion' on the census growing each decade (25% in 2011). This move is worthy of further exploration and study separately.

Sentence in the introduction revised from "In present day secular societies, interest in holistic care and mental wellbeing is growing." to "Within the NHS, interest in holistic care and mental wellbeing is growing."

Design- The authors states this is a qual study but they ask 3 quant questions so mixed methods.

This was a large survey, only part of which is addressed in this paper, and thank you for highlighting when this was confusing. This sentence was revised to: "A mixed methods online survey was developed and used to explore GP definitions of the term 'spiritual health', attitudes towards the topic, and views on discussing spiritual health with their patients."

Please identify how was the tool developed? I am unsure why the bracketed statement re tools to discuss consultations is here.

This survey was larger than that reported here, and the rest of the survey addressed tools for discussing spiritual health in primary care.

Why were the 3 quant questions asked and how do they link to the research question? Abstract reporting differs to design aim.

The questions for self rating "I am a spiritual person; I am a religious person and I am spiritually healthy" were asked to explore whether, before asking "what does the term 'spiritual health mean to you?" the participants had a working definition they were using to allow self rating, and act as an introduction. These questions also explored whether GPs consider being religious, and being spiritual, as synonymous. The literature regarding discussion of spiritual health in primary care emphasises the importance of self-awareness of our own spiritual health. However, the statement 'I am spiritually healthy' and the ratings given, does not link to the research question reported here, and is better reported with the findings on discussing spiritual health within the consultation, and so has been removed from this paper, for reporting elsewhere.

To clarify the questions asked, and how they relate to the research question, the rewritten paragraph now reads:

"A mixed methods online survey was developed and used to explore GP definitions of the term 'spiritual health', attitudes towards the topic, and views on discussing spiritual health with their patients. (Data on the use of structured tools to support discussions within consultations are reported elsewhere). Information collected from the participants included demographic characteristics (sex, ethnicity and religion); geographical region and years of work experience. Participants were asked to rate themselves on a five point Likert scale, in response to two statements: 1. I am a spiritual person and 2. I am a religious person. Spiritual health was deliberately undefined in the survey; GPs were asked 'what does the term spiritual health mean to you?' and invited to share any other thoughts on this topic, prompted by five patient vignettes."

Where is reporting of ethical approval.

The reporting of ethical approval is after the funding statement: "Ethical approval: Ethics approval was sought and obtained from Newcastle University on 27/2/2019"

Analysis- Which methodology did you utilise?

Thematic analysis was used, based on a priori themes from the literature shown in Box 1.

unclear from design if only 1 qual question was asked.

Thank you for highlighting the need for clarity on the questions. As mentioned in the revised paragraph above, this has been clarified to: "GPs were asked 'what does the term spiritual health mean to you?' and invited to share any other thoughts on this topic, prompted by five patient vignettes."

What does the author mean by deviant, explain.

Deviant was used as a term to mean analysis of cases which appear to fall at either end of a spectrum, or extremes of opinion, or contradictory evidence, to ensure the breadth of opinions had been understood by the researcher and included. In view of the connotations of the term 'deviant', this has been replaced by 'outlying' to clarify the meaning: "Outlying cases were examined, to identify insights from respondents most and least comfortable with the topic. "

Justify manual coding.

The data were small enough to allow it.

How did the researchers limit their bias?

In this qualitative work, the authors do bring their own perspectives to the research, and did reflect on this. Author 1 coded the data from the GP participant, and author 1's demographics reflect the largest group of GPs in England according to GMC records. This is detailed in the text: "The researchers were cognisant of their own identities, cultures and religious backgrounds in their approach to the topic. Feedback was sought from GPs from other backgrounds in the survey development. Authors 1 and 3 are practicing GPs."

Results- I think there needs to be clarity as to why the GPs were asked primarily about whether they were spiritual/religious or spiritually healthy.

The primary research question reported here is where the GPs were asked "what does the term spiritual health mean to you?" as well as "any comments?" after the five vignettes. The demographic information is given as an introduction to the results, and hasn't been amended. The two quantitative questions "I am a religious person" and "I am a spiritual person" were included to allow exploration of how GPs understand those two concepts, and whether they overlap, or are distinct. The authors feel that the 21% who were spiritual but not religious present an important finding in terms of how GPs understand the overlap between the concepts of spirituality and religiosity.

Rephrase line 50, of the 85 respondents xxxx

Thank you, this wasn't a clear sentence, rephrased to "All respondents who agreed they were religious also agreed they were spiritual people."

How were the a priori themes developed, expand please for clarity.

The a priori themes were developed by extensive reading and review of the current literature on how doctors should approach the topic of spiritual health in the consultation. The text was modified to "based upon a priori themes developed by immersion in the literature".

How many participants said dread of the afterlife- line 38, p8, if an outlier response please identify this.

This was one respondent, and on closer examination, was from Northern Ireland and should have been excluded. (The data were checked again that the three other excluded responses had not been included.) The sentence now reads: "Participants referred to the health of a soul, or inner self, which could be either peaceful or disrupted with concerns, conflict, or guilt"

Justify theme of meaningless - only 1 line included.

When reporting the themes, it would have been disingenuous to omit those who said the term 'spiritual health' had no meaning to them at all. These participants did not expand on their opinion, unfortunately, giving one or two word answers. This theme has been adjusted to "Spiritual health as a meaningless term" to clarify that it isn't analogous to nihilism, but that the term itself is meaningless.

In general the themes could benefit from some revision. When comments are made for example transcendence "child of God" identify if these are minority comments

Some participants referred to a God, but most did not, within their definitions.

Numbers of participants have been added to the text to contextualise these comments- "Some participants were clear that, for them, this was part of an accepted religious framework, for example "a relationship with Jesus" (one participant) or "myself as a Child of God" (18 participants mentioning God)."

As above, the theme of 'meaningless' has been clarified to "Spiritual Health as a Meaningless Term".

p10 line 50, why were data mapped to appleby, expand

The data were mapped to Appleby et al's previous qualitative work on GP attitudes to discussing spiritual health in primary care, in Scotland. Appleby et al had organised their data effectively, from a population similar to that under investigation here, therefore to build the evidence base it was felt to be useful to analyse this data similarly, and explore whether this population had similar findings to the Scottish GPs. The paragraph "Attitudes to spiritual care" was rewritten to "Respondents were invited to add any other thoughts on spiritual health and care in general through unprompted free text boxes asking "any comments". A third of participants contributed their views, and data relating to understanding and definition of spiritual health were mapped to an existing framework developed by Appleby and colleagues in Table 2. This is derived from a small qualitative survey of 19 Scottish GPs³ which was then used by the authors with a larger group of Scottish GPs in a quantitative study⁴, and presents a categorisation of the spectrum of attitudes to spiritual care already used in a UK context. The perceived need for cues by a large number of respondents, and the low numbers of respondents who fit into the 'rejecting' attitude, would suggest that most respondents fall in to the "guarded" and/or "pragmatic" attitudes identified."

Be consistent about UK GPs as abstract states England

Thank you, UK changed to 'English' at the start of the discussion.

Discussion line 15-26, pg 23 would be better placed in limitations section.

Unfortunately, the line numbers and page numbers differ in the document being revised, however the likely paragraph was relocated to the limitations section.

Line 29 what does this mean?

Apologies, due to the issue of the exact numbering of lines and pages not being the same, I am not sure exactly which line this refers to. "including a lack of meaning of the term 'spiritual' " has been added to clarify that the lack of meaning refers to the terms, rather than nihilism.

The discussion is very weak.

Thank you for your feedback on the need to expand the discussion, we are unclear as to the specific reasons why the discussion is weak and what we have omitted. However we have made the following amendments:

"In the UK" and "other" added to the paragraph below:

"We have been able to present an in-depth and nuanced description of GPs' understanding of spiritual health, with a study that is larger than previous work in the UK. We identified some views of spiritual health that were similar to those expressed in interviews with 19 Scottish GPs and a

quantitative survey of 87 GP trainers, including a lack of meaning of the term spiritual, links with humanity and the divine. 27 31 The breadth of views in our study was similar, enabling us to map responses to a framework based on Appleby's four descriptors of attitudes to spiritual care. Other work has focussed on a priori definitions of spiritual health.^{28 32-34} Most other literature originates from the USA, which may promote a definition that is appropriate only to that socio-cultural context." Why is Ellis et al (see below) not discussed. Ellis M, Vinson D & Ewigman B (1999) Addressing Spiritual Concerns of Patients: Family Physicians' Attitudes and Practices. *Journal of Family Practice* 48 105-9

Ellis M, Campbell J, Detwiler-Breidenbach A & Hubbard D (2002) What do Family Physicians think about Spirituality in Clinical Practice? *The Journal of Family Practice* 51 (3) 249-254

Ellis M & Campbell J (2004) Patient's Views about Discussing Spiritual Issues with Primary Care Physicians. *Southern Medical Journal* 97 (12) 1158-1163

The authors are aware of Ellis et al's work.. Ellis et al's work is based in the USA, with a USA population of patients and doctors, and therefore this work is not considered by the authors to be as comparable as work done in a similar population such as Appleby et al. The Ellis 2002 paper is about discussing spirituality with patients, rather than attitudes or defining spiritual health. While this is interesting in comparison to the other parts of this study, not reported here, it is of little relevance to this research question. The Ellis 1997 paper uses a quantitative scale, rather than a qualitative framework to assess spiritual wellbeing of doctors, rather than attitudes towards the concept, the scale had never been used in physicians before. Assing Hvidt et al have done some work in Denmark exploring the attitudes of Danish GPs to the 'existential dimension' of patient care, however they define this as separate to the spiritual dimension of health, and therefore it was not felt to be the best comparator here, although arguably the differences could be due to cultural interpretation of the terms. For similar reasons, comparisons were not made with Vermandere et al's work with Flemish GPs, as the data was brief, and possibly specific to the Flemish population and culture. It was felt that literature from the most similar culture was the best comparator. Pulchalski et al's consensus definition, while useful in developing the a priori themes, was not comparable to this work, as that definition was developed at a specialist conference into spiritual health, and this study gathered data from those with no interest, or in Appleby's terms 'rejecting' of the concept.

"Other work has focussed on a priori definitions of spiritual health,⁵⁻⁸ and while Puchalski et al's consensus definition⁹ gives a useful expert definition, it was previously unknown what definition English GPs were utilising in their work. Most other literature originates from the USA, which may promote a definition that is appropriate only to that socio-cultural context, whereas building on Appleby et al allows the UK evidence base to be developed. Assing Hvidt et al's work on 'existential health' in Denmark¹⁰ is an interesting comparator to this work, however their definition of existential health differs from that of 'spiritual health' here by English GPs, possibly due to cultural differences between the UK and Denmark." Was added to the discussion to strengthen the comparison between our work, and other literature on the topic.

Table 2 could be utilised much better.

To explain table 2's utility in the text, "as shown in table 2" added to the comparison with other literature, to put table 2 into the context of the discussion of Appleby et al's themes.

line 6, p9 when saying a majority of participants please add the number/percentage.

The percentage (63%) was added to the text.

line 36, p9 1 participant (remove s)

Thank you, this has been corrected.

Discussion on beneficial intervention should include some of Koenig' findings re positive impacts on health rather than initial link to job satisfaction. If adding this why is it linked?

The authors find Koenig's work on this topic informative. Koenig argues that spirituality and religion are interchangeable, and within his culture (USA) that may well be the case. His work looking at discussing spirituality and religion, and patient outcomes, is certainly interesting. As explained earlier, the introduction to this study is to highlight that there are potential benefits to patients and the GPs themselves of addressing the topic of spiritual health, but that the benefits are not being debated within this paper, the aim of this study is to explore the definitions of the term 'spiritual health' and attitudes of English GPs towards the topic. Therefore, the text has not been changed to include Koenig's work, as while it is interesting, it is not necessary for discussion of this question.

Reviewer: 2

Dr. Maryam Rassouli, Shahid Beheshti University of Medical Sciences

Comments to the Author:

The manuscript has been written about an important topic which should be considered as a priority in providing health care. However, there are some concerns in this regard which should be considered as follows:

Introduction

Please mention the exact aim of the research at the end of the introduction section.

Thank you, this was also highlighted by reviewer one, and so the research question has been made clearer, and the end of the introduction section has been rephrased to "A clear understanding of spiritual health will be essential, if it is to become a routine part of holistic care in general practice.

This study aims to explore how GPs understand and define spiritual health in their day to day practice, and build the evidence base regarding GP attitudes towards the topic."

Methods

The section needs more clarification.

How was sampling method? How was the sample size calculated?

Both convenience and snowball sampling were used. A formal sample size was not calculated, as this was an exploratory, qualitative study, in a population that can be quite difficult to recruit to research.

"both convenience and snowball sampling methods were used" was added to the text.

Is there any inclusion/exclusion criteria for entering GPs in the survey?

This sentence was rewritten for clarity: "Included were all practicing GPs in England, people who were temporarily out of practice (for example maternity leave) were encouraged to participate, but those who had retired and would not return to practice were excluded ."

The content of the survey and its structure is not clear. There was just a unique question regarding meaning of spiritual health, which participants should answer. It seems that the research approach is a qualitative one so I'm wondering why researchers didn't use interviews as well as focus group discussions in order to gathering data.

This data was part of a larger survey about the topic of spiritual health in the consultation in primary care. GPs are a very time poor population, and the topic of spiritual health can be quite stigmatised, and elicit polarised views. It was felt important that this population were given an opportunity to share their views that is anonymous, quick, easy and simple, outside of a group setting, where those with stronger opinions may dominate discussion. Explanation was added to the text, as suggested by the other reviewers, to explain that there was one main qualitative question, with free text comments also invited after five patient vignettes around spiritual health issues.

According to page 3, lines 56-57, data analyzed based on the provided definitions in the box 1.

Therefore, it seems that the researchers used "directed content analysis". Please explain which approach has been used for analyzing data according to this type of content analysis.

A deductive thematic analysis was used, using a priori themes from the literature, as described in the text.

It is recommended to mention what researchers did for confirming the trustworthiness of data. The data were approached in a systematic way, initially cutting out all responses and sorting into codes. Larger responses were split into multiple codes, if applicable. Author 1 developed these codes into themes, within the framework provided by the a priori themes from the literature. These themes were then explored with author 3, and discussed. This is covered in the 'Data Analysis' paragraph of the Method.

The reason for bringing the separated paragraph in page 4, line 28 (patient & public involvement) is not clear.

The text "Patient and Public Involvement was important to the development of the research and analysis, and the authors wish to acknowledge, and are grateful for, the contribution of Voice North." Has been added to explain that the authors value the contribution of patients and the public to this research.

It is recommended to describe number of participants who didn't answer the survey. Unfortunately, it is unknown how many GPs within England saw the survey invite, and did not respond.

It seems that the research has two parts that if the results of both parts are presented together, it can provide valuable information to readers. Due to the fact that the qualitative part was done only through questionnaires without interviewing with participants, it does not provide much information and therefore the discussion of the findings does not add anything to the existing knowledge.

We agree, thank you for making this point. It is unfortunate that this population are time poor, and that the topic can be a difficult one to address with GPs due to stigma, I agree that interviews could have provided rich data on the topic. However, the benefit of the survey is that a large population of GPs were sampled, gaining a variety of views on how GPs define spiritual health, and supporting Appleby et al's framework for exploring GP attitudes to the topic. It is planned that the second part of the study (the use of tools to discuss spiritual health with patients) will be published shortly. The strengths and limitations of this study are outlined in the text, so no additions were made. While interviews could give further information, the findings presented here do add to the existing knowledge due to the number and breadth of responses.

Reviewer: 3

Dr. Dirk Labuschagne, Rush University Medical Center

Comments to the Author:

This paper reports UK general practitioners' (GPs) understanding of spiritual health. Regulatory bodies in the UK require GPs to consider their patients' spiritual and religious needs. Yet, challenges around doing so and having no consensus on an accepted definition for spiritual health complicate this. The authors seek to explore GPs definitions of spiritual health as a first step in this regard.

The attempt to do so is well-written, with a comprehensive introduction and engagement with the literature, a clear methodology and proper summary of the results. There are a few suggestions/questions to further improve this study.

p. 0, line 35: This reviewer's reading of the study results doesn't agree with this statement that the GPs were asked about discussing spiritual health with patients. Another aspect of the survey may have asked about it (results not reported in this paper) but we read results asking GPs about their understanding of spiritual health. If this additional aim is part of the study (especially in regard to this

reviewer's comments below re: p. 4, line 17), then this should be included in the aim and discussion sections.

Thank you, the other reviewers have also highlighted this. The abstract has been rewritten and clarified for this part of the study: "A mixed methods online survey asked practicing GPs in England qualitative free text questions- "What does the term 'Spiritual Health' mean to you?" and "Any comments?" after five vignettes about discussing spiritual health with patients. These were subject to thematic analysis using a priori themes from the literature on GP definitions of spiritual health, and on attitudes towards the topic."

p. 0, line 45: Participants reported here belong in the result section; participants may just be qualified GPs in England (see p. 3, line 40)

The participants section rewritten to "Participants: 177 practicing GPs in England" and relocated to just above results section.

p. 3 line 9-10: Authors emphasize need for clear understanding of spiritual health and state aim of study as to "address this gap." It is not clear how reporting the GPs understanding of spiritual health is filling this gap, since similar to the wide variety of definitions already present in the literature (p. 2, line 58), GPs would probably have a wide variety of views, not clearly defined. At the very least, it seems to test whether GPs have a similar wide variety of definitions, thereby underlying the need for a consensus definition.

Thank you, this is a good point, we have sought to build on Appleby and other's work, and rather than a gap in definitions, our aim was to understand how non-expert GPs understand the term "spiritual health". This sentence was rewritten to "A clear understanding of spiritual health will be essential if it is to become a routine part of holistic care in general practice. This study aims to explore how GPs understand and define spiritual health in their day to day practice, and build the evidence base regarding GP attitudes towards the topic." And "However, while these definitions may be useful to those interested in the topic, or within academia, it is not known what working definition is being used by GPs in a clinical context." Was added.

p. 4, line 17: How did the free text space for additional comments result in the data that is presented in Table 2, p. 13? The data in Table 2 appears to come from a specific prompt, perhaps about their views regarding discussing spiritual health with patients? If so, does this necessitate adding this as a secondary aim of this study and updating the discussion, bringing into dialogue Table 2's findings with the relevant literature?

Thank you, yes, this needs to be clearer that the GPs were asked 'any comments' after five patient vignettes. Method rewritten to include: "GPs were asked 'what does the term spiritual health mean to you?' and invited to share any other thoughts on this topic, prompted by five patient vignettes."

p. 6, line 48-52: With the identification of "Self-Actualization and Meaning" as a theme, "Meaningless" isn't the best description here since it may be misleading. It denotes spiritual health as potentially being viewed as senseless or futile, rather than the more practical understanding that for some GPs spiritual health felt unclear or undefined.

We agree this is a good point, and also raised by reviewers one and two. The theme heading is changed to "Spiritual health as a meaningless term", and changed in discussion to "We identified some views of spiritual health that were similar to those expressed in interviews with 19 Scottish GPs and a quantitative survey of 87 GP trainers, including a lack of meaning of the term "spiritual" and links with humanity and the divine."

p. 8, line 7: It is not clear to this reviewer based on the study results how the meaning connected to spiritual health was distinct from religion. The authors didn't define their working definition for religion,

and in the themes identified several distinctly religious items emerged, such as punishment, afterlife, soul, Jesus, and child of God.

The authors have deliberately left the terms spiritual health, religion and spiritual open and undefined, for interpretation by each participant. The authors did find it interesting that a fifth of respondents considered themselves spiritual people, but not religious people, demonstrating these fifth at least think that spirituality is distinct from religion. A sentence has been added to the discussion saying “While some authors describe religion and spirituality as interchangeable¹¹, a fifth of the GP participants here described themselves as spiritual but not as religious, showing they believe these are different concepts”

p. 8, line 46: It is not a clear parallel to connect the lack of meaning as identified in the referenced study (where GPs were asked about their understanding of spirituality) and the present study where GPs left as undefined their definition for spiritual health.

Similarly to those in this study who stated the term spiritual health is unclear or meaningless, those in the Appleby study also expressed that spirituality is an unclear or meaningless concept. It is hoped that this is clarified by the addition described above where “including a lack of meaning of the term “spiritual”” was added. The sentence “The definitions given by English GPs reflects those given in the literature (box 1).” Is also added.

p. 15, line 13: The authors’ first names are listed, correct to Cobb, Puchalski and Rumbold
This has been corrected, thank you.

VERSION 2 – REVIEW

REVIEWER	Dirk Labuschagne Rush University Medical Center, Supportive Oncology
REVIEW RETURNED	18-May-2021
GENERAL COMMENTS	Thank you for this opportunity to review this revision. The authors have appropriately addressed the issues I've raised in my review.